# Effect of Cleansers on the Colour Stability of Zirconia Impregnated PMMA Bio-Nanocomposite

**DOI:** 10.3390/nano10091757

**Published:** 2020-09-06

**Authors:** Saleh Zidan, Nikolaos Silikas, Julfikar Haider, Julian Yates

**Affiliations:** 1Dentistry, School of Medical Sciences, University of Manchester, Manchester M13 9PL, UK; nikolaos.silikas@manchester.ac.uk (N.S.); julian.yates@manchester.ac.uk (J.Y.); 2Department of Dental Materials, Faculty of Dentistry, Sebha University, Sebha, Libya; 3Department of Engineering, Manchester Metropolitan University, Manchester M1 5GD, UK; j.haider@mmu.ac.uk

**Keywords:** denture, PMMA, zirconia (ZrO_2_), nanocomposite, colour stability

## Abstract

Exposure of denture base acrylic resins to the oral environment and storage media for extended periods of time results in colour change due to changes in the properties of the material. The purpose of this in vitro study was to assess the colour stability of high-impact heat-polymerized denture base acrylic resin (HI PMMA) impregnated with zirconia nanoparticles after storage in distilled water (DW) and denture cleaners such as Steradent (STD) and Milton (MIL) for 180 days. Ninety specimens of PMMA + Zirconia nanocomposite with varying nanoparticle concentrations (1.5 wt.%, 3.0 wt.%, 5.0 wt.%, 7.0 wt.% and 10 wt.%) were prepared with a diameter and thickness of 25 ± 1.0 mm × 2 ± 0.1 mm and divided into six groups, while each group was further divided into three subgroups: storage in DW (control), STD and MIL. Colour changes were measured with a Minolta Chroma Meter (Minolta, Osaka, Japan), and assessed using the CIE L*a*b* colorimetric system. Data were statistically analysed for colour change with Friedman’s Two-way and Kruskal-Wallis tests at a pre-set alpha value level of 0.05. The colour change (Δ*Ε*) exhibiting significant differences were found among all groups immersed in denture cleaners, and all values increased with time. According to the National Bureau of Standards, the control group displayed the lowest colour change value (Δ*Ε* = 1.22), and the highest value was for 10 wt.% ZrO_2_ while stored in MIL (Δ*Ε* = 6.07). The values of colour change for storage in water ranged from 0.49 (control) to 1.82 (10 wt.% ZrO_2_). The colour change value for the composite group containing 3 wt.% zirconia was clinically acceptable. However, high concentrations of denture cleaners should be avoided, and the shortest cleaning time is recommended to improve the clinical life of the nanocomposite denture base.

## 1. Introduction

Poly-methyl methacrylate (PMMA) acrylic resin has been accepted as the material of choice for constructing artificial denture bases since the beginning of the twentieth century [1]. PMMA resin has been reinforced with butadiene-styrene rubber to develop high-impact PMMA in an attempt to improve its physical and mechanical properties. These materials are provided in a powder-liquid form and processed in the same way as other conventional heat-polymerized methyl methacrylate resins [2]. Within the oral environment, the physical properties of PMMA undergo rapid changes with time and this influences the mechanical properties [3]. In relation to general aesthetics, in the long term, staining affects the colour, gloss surface and the shine of the denture, which are of concern to both patient and dentist [3].

Denture cleaners are commonly employed to remove stains and debris from denture surfaces and to prevent the formation of plaque or colonization of bacteria. A study was conducted by Cantore et al. with Periogen mouthwash cleaner to evaluate its antimicrobial activity using collected saliva samples. The study concluded that antimicrobial action of the cleaner in the mouth could resist plaque formation [4]. However, their daily use may impact the physical properties of the denture base resin, affecting its surface roughness, hardness, gloss or color [5]. In addition, a lack of information on using cleaning solutions correctly may affect the denture properties [6].

Several studies [7,8,9,10], have investigated the short and long-term effects of denture cleaners such as efferdent, chlorhexidine, sodium hypochlorite, peroxide and other commercial cleaning agents on the colour stability of different denture base resins. They found that the colour change (defined by Δ*Ε*) of the denture bases was influenced by the type of denture cleaners used, and Δ*Ε* values increased with an increase in duration in exposure to cleaning solutions. McNeme et al. found that denture cleaners (sodium hypochlorite) could cause bleaching of a denture base acrylic resin [11]. Polychronakis et al., also reported a significant decrease in the gloss and higher surface roughness with different denture base resin materials immersed in dental cleaning agents (Corega Extradent) [12].

Recent literature shows that the addition of nanoscale reinforcing agents with polymer materials produces improved mechanical and physical properties, creating a new class of nanocomposite. In dentistry, many attempts have been made to develop PMMA based nanocomposite with the addition of different nanosized fillers [13] and their physical and mechanical properties are evaluated. However, there is no study in the literature investigating the colour stability of High Impact (HI) heat-cured denture base resins reinforced with zirconia nanoparticles. 

The aim of this study was to evaluate the effect of adding zirconia nanoparticles at different concentrations to high-impact PMMA on colour change when stored in distilled water (DW) and two different media of denture cleaners Steradent (STD) and Milton (MIL) for up to 180 days. The hypotheses to be tested were that there would be no significant change in colour stability for HI PMMA nanocomposite experimental groups compared to the pure HI PMMA (control group).

## 2. Materials and Experimental Method

### 2.1. Materials

A commercially available PMMA acrylic resin as matrix and zirconia (ZrO_2_) nanoparticles as filler were chosen for fabricating denture base nanocomposites [14]. The details of the denture base materials and cleaning solutions used to conduct the colour change experiments are shown in Table 1.

### 2.2. Preparation of Specimens

Zirconia nanoparticle surfaces was treated with 7 wt.% silane coupling agent (3-trimethoxysilyl propyl methacrylate; Sigma-Aldrich (St. Louis, MO, USA) as detailed in a previous study [14] to facilitate the bonding between the nanoparticles and the matrix by its alkoxysilane groups reacting with the filler and itself, and with the methacrylate functional group within the resin [15,16]. According to the study [14], the most appropriate weight percentages of zirconia nanoparticles were used in this study: 0% (G1, control), 1.5 wt.% (G2), 3.0 wt.% (G3), 5.0 wt.% (G4), 7.0 wt.% (G5), and 10.0 wt.% (G6). The silane-treated zirconia was mixed with acrylic resin according to the manufacturer’s instructions until the mixture reached a dough-like stage, which was packed into a mould. The mould was kept under a pressure of 15 MPa and immersed in curing bath for 6 h at a temperature of 95 °C, to complete the polymerization process. The surface preparation started with grinding by a diamond bur and then with a tungsten carbide bur (Dental Sky, Kent, UK) at 1500 rpm to remove any excess acrylic on the surface. The first step of all surface polishing was carried out by a lathe bristle brush using slurry of pumice at the same speed as grinding for 1 min. The specimen surfaces were subjected to a second stage polishing with a muslin buff wheel using primary polishing compound (Chaperlin & Jacobs Ltd., Sutton, UK) at the same speed for 1 min. The third stage of polishing was carried out with a muslin buff wheel using fine polishing compound (secondary) at a slower speed of 500 rpm. Specimens were then cleaned with water and then dried in preparation for colour measurement.

### 2.3. Colour Measurement Procedure

Specimens were immersed into two cleaning solutions (Steradent (STD) and Milton (MIL)) and DW to measure the colour change. The cleaning solutions were prepared by immersing one STD tablet (3.184 g) into 250 mL of DW and 20 mL of MIL into 250 mL of DW. Both solutions were stirred for 30 s until the solutes were completely dissolved.

Minolta Chroma Meter CR-221 (Minolta, Osaka, Japan), a tristimulus colour analyser was used for measuring the colour and colour change [17,18]. The dimensions of the specimens were 25 mm diameter × 2 mm thickness to meet the demands of the measuring instrument. A total of 90 specimens were fabricated with fifteen specimens for each group (6 groups). Every group was divided in to three sub-groups (A, B and C). All specimens were immersed in DW and stored in an incubator at 37 ± 1 °C for 24 h. The specimens were dried with a tissue paper before the colour measurement and placed into the holder of the measuring head. During the measurement, light was shone on the specimen at an angle of 45° and the reflected light was analysed [18]. Three measurements were recorded for each specimen as the baseline measurements. Then, five specimens from subgroup A were immersed in 40 mL DW to serve as control, five specimens from subgroup B were immersed in 40 mL STD solution, and the remaining five specimens from subgroup C were immersed into 40 mL MIL solution. The colour values after baseline measurements were collected at 7, 14, 21, 30, 60, 90, 150 and 180 days. 

The Minolta Chroma Meter CR-221 was calibrated before each measurement using the white calibration plates, according to the manufacturer’s instructions. The colour differences were assessed by the International Commission on Illumination (CIE) L*a*b* scale, where *L** is the lightness, *a** defines the colour of the sample on red-green and *b** on yellow-blue axes. The colour change (Δ*Ε*) values were calculated by using Equation (1) [18]:Δ*Ε* = [(Δ*L**)^2^ + (Δ*a**)^2^ + (Δ*b**)^2^]^½^(1)
where, Δ*L**, Δ*a** and Δ*b** represent the differences between the *L**, *a** and *b** values of the baseline coordinates and those measured after immersion [9,19].

Level of colour change (NBS) was quantified according to the American National Bureau of Standards (Table 2). The NBS values were calculated using Equation (2) [9].
NBS = Δ*Ε* × 0.92(2)

### 2.4. Statistical Analyses

The calculated data of colour change were statistically analysed using SPSS statistics version 23 (IBM, New York, NY, USA). The Shapiro–Wilk test showed that the data for the colour change (Δ*Ε*) were not normally distributed. Therefore, non-parametric tests were used to analyse the results on three factors (time, solutions and groups). The Friedman’s Two-way test with related samples was analysed for the effect of time followed by the Kruskal–Wallis test with independent samples analysed variances on the solutions and groups. The data were checked to see whether there were significant differences among the three factors at a pre-set alpha value of 0.05.

## 3. Results

The colour changes of HI PMMA nanocomposite denture specimens after 180 days of immersion in two different cleansers and DW are presented in Figure 1. Baseline of (CIE) L*a*b* colour measurement values (mean and standard deviations) of the six experimental groups after 24 h immersion in DW are listed in Table 3. The median values, Interquartile Range (IQR) and colour expressed in NBS unit, for the colour change (Δ*E*) measurements after 7, 30, 60, 90, 150 and 180 days are shown in Table 4. 

In general, for all groups and after all immersion durations, there was a difference in colour change between the samples immersed in DW and the other solutions (Table 5). In the majority of cases, the difference was not significant. However, significant differences were found for only three groups. Firstly, for the group containing 1.5 wt.% ZrO_2_, after 180 days there was a significant increase in colour change when compared between immersion under DW and STD. Secondly, for the group containing 3.0 wt.% ZrO_2_, after 150 days there was a significant increase in colour change when compared between immersion under DW and MIL. Thirdly, for the group containing 7.0 wt.% ZrO_2_, after all immersion intervals, there was a significant increase in colour change when compared between immersion under DW and STD, and immersion under DW and MIL.

Again, an increasing trend of colour change was observed between 7 days and 180 days for any specimen group immersed in any individual cleaner, but not significantly different (*p* > 0.05). 

Significant differences (*p* < 0.05) were found in colour change (Δ*E*) between all groups containing zirconia and the control group after immersing in STD cleaner at 90 and 150 days. However, for any particular immersion duration in DW and MIL solution, there was no significant colour difference among the groups.

The Friedman’s Two-way test with related samples and the Kruskal-Wallis test with independent samples showed that within each experimental group, there was significant differences (*p* < 0.05) between distilled water and the two cleaners (STD and MIL) over time. In the control group (0% ZrO_2_) significant differences in colour change (Δ*E*) between specimens stored in DW and MIL cleaner were observed at 150 days. Specimens containing 1.5 wt.% of ZrO_2_ immersed in STD showed significant differences in colour change (Δ*E*) at 180 days compared to DW. The group containing 3 wt.% ZrO_2_ immersed in MIL also showed significant increases in colour change (Δ*E*) compared to DW at 150 days. Significant differences in the colour change (Δ*E*) were also found in specimens containing 7 wt.% ZrO_2_ immersed in MIL compared to DW at all intervals. However, in groups containing 5 wt.% and 10 wt.% ZrO_2_, there were no significant differences in colour change (Δ*E*) between two the cleaners and distilled water (Table 5).

The values of colour change (Δ*E*) at 180 days were expressed in NBS values as listed in Table 6 and presented in Figure 2. Specimens in groups containing 1.5 wt.% and 3 wt.% of ZrO_2_ immersed in DW demonstrated “slight change” compared to the control group, while the other groups (5 wt.%, 7 wt.% and 10 wt.%) showed “noticeable change”. However, the control group also demonstrated “slight change” following immersion in STD and MIL cleaners. All ZrO_2_ containing specimens in both MIL and STD solution showed at least noticeable change or higher. 

It was interesting to notice that colour change was strongly associated with whitening (*L**) between control group and other groups containing zirconia in contrast to redness/greenness (*a**) and yellowness/blueness (*b**) values as shown in Table 3. In addition, as the concentration of zirconia was increased, the whitening effect also increased and this is visually evident in Figure 1. 

## 4. Discussion

In this study, there was a significant difference in the colour stability of HI PMMA nanocomposites with different concentrations of ZrO_2_ when stored in different storage media, therefore the hypothesis was rejected. The colour changes of all experimental groups increased as the storage time increased. In addition, specimens stored in STD and MIL cleaners showed more colour change than those stored in DW (Figure 1). The control group exhibited less colour change (ΔE) after 180 days storage in cleaners, whereas groups containing 1.5 wt.% and 3 wt.% of ZrO_2_ demonstrated clinically acceptable colour change, the group containing 5 wt.% of ZrO_2_ showed colour change slightly above the clinically acceptable value, and the groups containing 7 wt.% and 10 wt.% of ZrO_2_ showed colour change above the clinically accepted values (Figure 2).

It was interesting to investigate the difference in colour change (ΔE) as result of the addition zirconia nanoparticles to HI heat-polymerized denture base acrylic resin. When specimens stored in distilled water at 7 days were compared with baseline data, the finding showed that there were slight changes in colour values between the groups containing zirconia and the control group (without ZrO_2_) (Table 4). As the concentration of ZrO_2_ increased, the colour change increased, however, these changes were clinically acceptable according to NBS [20,21]. Therefore, the combined effect of the cleaning solutions and concentration of ZrO_2_ nanoparticles could be accounted for the colour change in the nanocomposite specimens. 

No identical studies are available in the literature for direct comparison with the present study. However, a similar study was reported by Davi et al., who evaluated the colour stability of microwave polymerized acrylic resin immersion in different percentages of sodium hypochlorite (0.5% and 1%), Clorox/Calgon and water after 180 days. They reported significant colour change with 1% of sodium hypochlorite cleaner, whereas Clorox/Calgon and 0.5% of sodium hypochlorite produced no whitening effect. It was explained that the concentration of sodium hypochlorite and length of immersion could have affected the colour change [7]. Hong et al., found that colour change in acrylic resins increased with immersion time and was caused by monomer leaching out and water being absorbed [9]. Another study by Peracini et al. investigated the colour stability of heat-cured denture base acrylic resins following immersion in effervescent denture cleaners (Corega tabs and Bony Plus) over 180 days. Although, a higher colour change (Δ*E*) was measured for the both cleaners, the differences were clinically insignificant [6]. In addition, Goiato et al., reported significant colour change (Δ*E*) of denture base acrylic resins after immersing for 60 days in Efferdent, 4% chlorhexidine and 1% hypochlorite; however, all values were within clinically acceptable parameters [8]. The findings of the current study are in agreement with the aforementioned studies as the colour change values obtained from the control group storage in MIL (sodium hypochlorite 2%) and STD after 180 days provided similar results. However, groups containing ZrO_2_ showed higher colour change (Δ*E*) values for both cleaners. 

Polyzois et al. evaluated the effect of water, peroxide and hypochlorite (5.25%) cleaners on the gloss, colour, and sorption of pink acetal and thermoplastic acrylic resins after 100 days immersion. They found the highest colour change (Δ*E*) for the resin after storage in water was 7.64, whereas in hypochlorite it was 26.54, which was clinically unacceptable [10]. This is in disagreement with the present study, where the highest value of colour change was measured as 1.82 (storage in water) for the group containing 10 wt.% ZrO_2_ and was clinically acceptable. However, storage in MIL for 180 days resulted in a colour change value (Δ*E*) of 6.07, which was clinically unacceptable and in agreement with Polyzois’ study. 

Colour and gloss are important determiners of the aesthetics of dentures, whereby changes in colour and gloss are indicators of the degradation of the denture base material [12]. The increase in the colour change (Δ*E*) values suggested loss of surface gloss [22]. It has been reported that a smooth surface prevents the formation of discolouring films, affects plaque formation and facilitates its removal [10]. The composition of the acrylic resin and built-up substances can also influence the colour stability, especially if the surface roughness is higher [23]. Polychronakis et al. evaluated the influence of denture cleaners on colour, surface roughness and gloss of nylon and heat polymerized acrylic denture base resins before and after 30 days of immersion [12]. The finding showed that the surface roughness, gloss and colour were affected in both types of dentures. This is in agreement with our finding that groups containing 5 wt.%, 7 wt.% and 10 wt.% of ZrO_2_ (Figure 2) showed higher colour change (Δ*E*) after 180 days storage in cleaners compared to the control group. Even though gloss was not measured in this study, it could be assumed that this change in colour could affect the gloss layer on the specimen surfaces. 

Goiato et al. found that the release of monomer during storage could interact with the glaze layer and cause increased colour change [23]. In addition, residual monomer can react with pigments inside the polymer, causing further colour changes [3]. Goiato et al., evaluated the effect of the addition of nanoparticle pigmentation (3% and 7%) on the colour stability of acrylic resin immersed in water [24]. They reported that the addition of pigments reduced colour change (Δ*E*) and that the lowest colour change value was found with samples of 7% pigmentation. In our study, the control group showed lowest colour change (Δ*E*) in all three-storage media in contrast to the groups containing ZrO_2_. This could be related to change of pigment in the control group as an increase in ZrO_2_ concentration could gradually replace the pigment and increase the colour change. 

The effect of high colour change appeared as whitening or bleaching in some areas on specimen surfaces (Figure 1). In the current study, a possible explanation for these results, especially when stored in denture cleaners over time, might be related to sodium hypochlorite in the composition of MIL, which was reported to cause the bleaching of acrylic resin in the study by Goiate et al [8]. Moreover, Me Neme et al. assessed the effect of sodium hypochlorite 1% and other cleaners over time from 15 min to 72 h on the colour stability of denture base acrylic resins. They reported colour change (Δ*E*) after 2 h, and bleaching was noticed after immersing for 72 h [11]. Several authors reported that peroxide-type denture cleaners include an effervescent component such as sodium perborate or sodium bicarbonate. When effervescent tablets dissolve in water, the sodium perborate readily decomposes to form an alkaline peroxide solution [6,9]. The alkaline peroxide solution reduces surface tension and agents such as sodium perborate release oxygen from the solution [25]. Through reduction, these free radicals convert larger molecules into smaller ones [26], which influences both mechanical and chemical cleaning [25]. Other researchers mentioned that denture cleaners could affect denture acrylic resins after a period of storage and caused whitening or bleaching, loss of soluble components and monomer leaching out [9,27,28], while water absorption affected their physical properties [29]. This could be an explanation for the results found in this investigation, where the bleaching action was caused by peroxide content in STD and sodium hypochlorite in MIL. MIL showed higher colour change (Δ*E*) values than STD. It can also be noticed by visual inspection (Figure 1) that the most striking colour change was caused by the addition of ZrO_2_ rather than the immersion in denture cleaners. With higher zirconia content, a very pale pink/white colour was observed. This could be due to the addition of white colour zirconia into the pink PMMA.

To overcome the limitations of the current in-vitro study, further investigation should be designed to evaluate gloss for HI PMMA nanocomposite denture base. The specimen size should be increased to 10 specimens per group in order to obtain a better statistical distribution. To overcome the colour change of HI PMMA denture base materials (pale pink/white) due to the addition zirconia, natural biocompatible colour pigments can be added to improve the aesthetic. Even though the specimens were polished, the surfaces did not appear quite flat particularly in the experimental groups (Figure 1) possibly due to some small-scale surface deformation caused by immersion under denture cleaners for 180 days.

Biomaterials are artificial materials used to renovate or restore function of human body replacing a damaged part in order improve the quality of life [30]. In vitro studies are carried out on the biomaterials or scaffolds as a reparative model to apply safely in human models. Marrazzo et al. studied the effects of human platelet lysate (PL) on the repairing properties of dental pulp stem cells (DPSCs) suggested that the PL was a suitable alternative to fetal bovine serum (FBS) that could be used for the expansion and differentiation of DPSCs in vitro [31]. In this study, it has been demonstrated that the HI PMMA denture base impregnated with up to 3 wt.% of zirconia is capable of maintaining clinically acceptable colour stability for a long period of time (180 days) in DW and denture cleaners. The commercial denture cleaners used in this study can prevent plaque formation by effectively preventing bacterial colonization. Therefore, this study simulated the clinical conditions of the oral environment with which the new denture base nanobiocomposite would come into contact in day to day use. In our previous study, it was found that with 3–5 wt.% zirconia in HI PMMA can significantly improve mechanical properties [14]. This nanocomposite will have the potential to replace traditional PMMA and increase the clinical life of artificial dentures.

## 5. Conclusions

Within the limitations of this in vitro study, the following conclusions can be drawn:

Colour changes were affected by denture cleaners Steradent (STD) and Milton (MIL) over time for all experimental groups, the highest colour changes observed in the groups with 7 wt.% and 10 wt.% of ZrO_2_ immersed in MIL, followed by STD after 180 days. 

Based on the NBS unit, all groups in DW, and groups including control and 1.5 wt.% and 3 wt.% of ZrO_2_ in denture cleaners, were clinically acceptable. 

However, the groups with 5 wt.%, 7 wt.% and 10 wt.% ZrO_2_ were unacceptable for cleaning with STD and MIL.

The results demonstrated that the colour stability of the HI PMMA-Zirconia nanocomposite was influenced by the type denture cleansers used. 

For clinical application, the dental technicians must be aware of the potential colour change when zirconia is added to PMMA greater than 3 wt.% in combination with certain denture cleaners.

## Figures and Tables

**Figure 1 nanomaterials-10-01757-f001:**
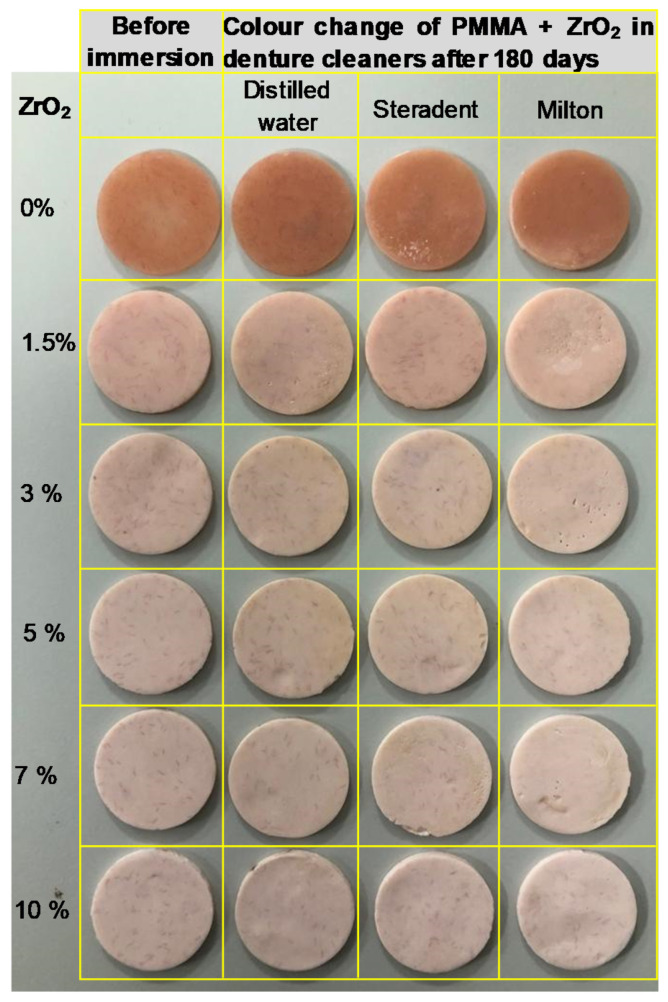
Photographs showing colour changes in the poly-methyl methacrylate (PMMA) + ZrO_2_ samples before and after immersion in different solutions for up to 180 days.

**Figure 2 nanomaterials-10-01757-f002:**
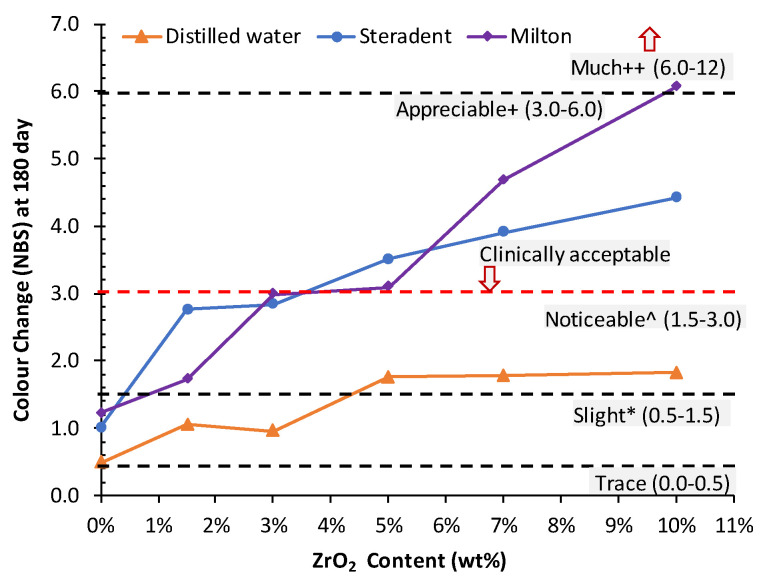
Line graph showing the colour change according to NBS unit system in specimens immersed in DW, Steradent (STD) and Milton (MIL) for 180 days.

**Table 1 nanomaterials-10-01757-t001:** Materials and denture cleaners used in this study.

Materials	Trade Name	Manufacturer	Lot. Number
High impact heat-curing acrylic denture base resin	HI Metrocryl	Metrodent Limited, Huddersfield, UK	Powder (22828) Liquid (103/4)
Yttria-stabilized zirconium oxide	Zirconium oxide	Sky Spring Nano Materials, Inc., Houston, TX, USA	8522-120315
Sodium bicarbonate, potassium carbonate peroxide, sodium sulfate, citric acid	Steradent	Reckitt Benckiser Healthcare, Dansom Lane, Limited, Hull, UK	0378493
Sodium hypochlorite	Milton	Laboratoire, Rivadis, Louzy, France	7227-430939

**Table 2 nanomaterials-10-01757-t002:** Critical marks of colour difference according to the National Bureau of Standards [9,20,21].

Textile Terms (NBS Unit)	Critical Marks of Colour Difference	
0.0–0.5	Trace	Extremely slight change
0.5–1.5	Slight	Slight change
1.5–3.0	Noticeable	Perceivable
3.0–6.0	Appreciable	Marked change
6.0–12.0	Much	Extremely marked change
12.0 or more	Very much	Change to other colour

**Table 3 nanomaterials-10-01757-t003:** Mean and Standard Deviations (SD) of baseline colour measurements after 24 h’s immersion in distilled water.

Experimental Groups	*L** Mean (SD)	*a** Mean (SD)	*b** Mean (SD)
G1 (Control)	8.98 (0.64)	0.342 (0.00)	0.331 (0.00)
G2 (1.5%)	33.96 (2.90)	0.346 (0.00)	0.342 (0.00)
G3 (3.0%)	44.67 (3.59)	0.344 (0.00)	0.343 (0.00)
G4 (5.0%)	54.05 (2.38)	0.345 (0.00)	0.343 (0.00)
G5 (7.0%)	56.53 (2.42)	0.342 (0.00)	0.342 (0.00)
G6 (10.0%)	60.19 (3.70)	0.339 (0.00)	0.340 (0.00)

**Table 4 nanomaterials-10-01757-t004:** Median values and Interquartile Range (IQR) of colour change and National Bureau of Standards (NBS) unit for specimens (control 0 wt.%, 1.5 wt.%, 3 wt.%, 5 wt.%, 7 wt.% and 10 wt.% of zirconia) immersed in distilled water, Steradent and Milton over time.

Experimental Groups	Cleaners	Day 7 Δ*E* IQR NBS	Day 30 Δ*E* IQR NBS	Day 60 Δ*E* IQR NBS	Day 90 Δ*E* IQR NBS	Day 150 Δ*E* IQR NBS	Day 180 Δ*E* IQR NBS
G1 (Control 0%)	Distilled water	0.20 (0.80) 0.18	0.33 (0.95) 0.30	0.51 (0.76) 0.46	0.49 (0.54) 0.45	0.52 (0.84) 0.47	0.54 (1.06) 0.49
	Steradent	0.31 (0.85) 0.28	0.49 (0.85) 0.45	0.52 (0.87) 0.47	0.49 (0.97) 0.45	0.75 (0.47) 0.69	1.10 (0.82) 1.01
	Milton	0.91 (0.53) 0.83	0.91 (0.92) 0.83	0.91 (0.99) 0.83	0.97 (0.74) 0.89	1.41 (0.45) 1.29	1.33 (1.43) 1.22
G2 (1.5%)	Distilled water	0.43 (0.88) 0.39	0.44 (1.21) 0.40	0.50 (1.21) 0.46	0.54 (1.37) 0.49	0.63 (2.85) 0.57	1.15 (3.63) 1.05
	Steradent	1.62 (7.38) 1.49	1.62 (7.63) 1.49	1.62 (7.65) 1.49	2.17 (3.57) 1.99	2.76 (6.57) 2.53	3.01 (5.79) 2.76
	Milton	1.78 (7.67) 1.63	1.78 (7.45) 1.63	1.78 (7.54) 1.63	1.55 (8.25) 1.42	1.46 (7.55) 1.34	1.89 (9.30) 1.73
G3 (3%)	Distilled water	0.54 (2.26) 0.50	0.56 (2.26) 0.51	0.56 (2.15) 0.51	0.57 (1.83) 0.52	0.64 (1.76) 0.58	1.05 (1.98) 0.96
	Steradent	2.39 (4.12) 2.19	2.39 (3.87) 2.19	2.45 (3.89) 2.25	2.81 (3.33) 2.58	2.89 (3.10) 2.65	3.09 (3.56) 2.84
	Milton	1.15 (5.97) 1.05	1.24 (5.98) 1.14	1.46 (5.90) 1.34	2.84 (3.50) 2.61	3.05 (3.27) 2.80	3.25 (3.70) 2.99
G4 (5%)	Distilled water	0.64 (2.72) 0.59	0.69 (2.57) 0.63	0.69 (2.57) 0.63	1.88 (3.72) 1.72	1.87 (3.29) 1.72	1.92 (2.55) 1.76
	Steradent	2.26 (4.40) 2.07	2.43 (4.35) 2.23	2.43 (4.55) 2.23	2.83 (2.77) 2.60	2.95 (3.64) 2.71	3.82 (3.53) 3.51
	Milton	3.51 (4.54) 3.22	3.51 (4.44) 3.22	3.51 (4.39) 3.22	3.96 (2.11) 3.64	3.98 (2.89) 3.66	3.37 (3.90) 3.14
G5 (7%)	Distilled water	0.68 (1.60) 0.63	0.97 (1.51) 0.89	1.26 (1.59) 1.15	1.33 (0.62) 1.22	1.54 (2.46) 1.41	1.94 (1.71) 1.78
	Steradent	3.85 (2.82) 3.54	3.85 (2.57) 3.54	3.85 (2.55) 3.54	3.93 (3.32) 3.61	4.46 (3.23) 4.10	4.26 (2.95) 3.91
	Milton	4.21 (5.48) 3.87	4.25 (5.38) 3.91	4.25 (5.08) 3.91	4.19 (3.25) 3.85	4.70 (4.67) 4.32	5.10 (4.41) 4.69
G6 (10%)	Distilled water	0.70 (3.32) 0.64	0.74 (3.31) 0.68	0.78 (3.41) 0.71	1.23 (2.39) 1.13	1.92 (2.19) 1.76	1.98 (3.84) 1.82
	Steradent	2.46 (7.31) 2.26	2.46 (7.50) 2.26	2.46 (7.57) 2.26	2.36 (6.09) 2.17	4.83 (6.62) 4.44	4.81 (7.62) 4.42
	Milton	3.66 (7.23) 3.36	3.66 (7.48) 3.36	3.66 (7.58) 3.36	3.20 (6.07) 2.94	5.36 (6.01) 4.93	6.60 (6.71) 6.07

**Table 5 nanomaterials-10-01757-t005:** The median colour change values and statistical differences of specimens following immersion in STD and MIL compared to DW for 180 days.

Experimental Group	Period (Days)	NBS Values after 180 Days in Cleaners		
		Distilled Water	Steradent	Militon
G1 (Control 0%)	7	0.20^A, a^	0.31^A, a^	0.91^A, a^
	30	0.33^A, a^	0.49^A, a^	0.91^A, a^
	60	0.51^A, a^	0.52^A, a^	0.91^A, a^
	90	0.49^A, a^	0.49^A, ϕ, a^	0.97^A, a^
	150	0.52^A, a^	0.75^A, ϕ, a, b^	1.41^A, b^
	180	0.54^A, a^	1.10^A, a^	1.33^A, a^
G2 (1.5%)	7	0.43^A, a^	1.62^A, a^	1.78^A, a^
	30	0.44^A, a^	1.62^A, a^	1.78^A, a^
	60	0.50^A, a^	1.62^A, a^	1.78^A, a^
	90	0.54^A, a^	2.17^A, Ω, a^	1.55^A, a^
	150	0.63^A, a^	2.76^A, Ω, a^	1.46^A, a^
	180	1.15^A, a^	3.01^A, b^	1.89^A, a, b^
G3 (3%)	7	0.54^A, a^	2.39^A, a^	1.15^A, a^
	30	0.56^A, a^	2.39^A, a^	1.24^A, a^
	60	0.56^A, a^	2.45^A, a^	1.46^A, a^
	90	0.57^A, a^	2.81^A, Ω, a^	2.84^A, a^
	150	0.64^A a^	2.89^A, Ω, a, b^	3.05^A, b^
	180	1.05^A, a^	3.09^A, a^	3.25^A, a^
G4 (5%)	7	0.64^A, a^	2.26^A, a^	3.51^A, a^
	30	0.69^A a^	2.43^A, a^	3.51^A, a^
	60	0.69^A, a^	2.43^A, a^	3.51^A, a^
	90	1.88^A, a^	2.83^A, Ω, a^	3.96^A, a^
	150	1.87^A, a^	2.95^A, Ω, a^	3.98^A, a^
	180	1.92^A, a^	3.02^A, a^	3.37^A, a^
G5 (7%)	7	0.68^A, a^	3.85^A, b^	4.21^A, b^
	30	0.97^A, a^	3.85^A, b^	4.25^A, b^
	60	1.26^A, a^	3.85^A, b^	4.25^A, b^
	90	1.33^A, a^	3.93^A, Ω, b^	4.19^A, b^
	150	1.54^A a^	4.46^A, Ω, b^	4.70^A, b^
	180	1.94^A a^	4.26^A, b^	5.10^A, b^
G6 (10%)	7	0.70^A, a^	2.46^A, a^	3.66^A, a^
	30	0.74^A, a^	2.46^A, a^	3.66^A, a^
	60	0.78^A, a^	2.46^A, a^	3.66^A, a^
	90	1.23^A, a^	2.36^A, Ω, a^	3.20^A, a^
	150	1.92^A, a^	4.83^A, Ω, a^	5.36^A, a^
	180	1.98^A, a^	4.81^A, a^	6.60^A, a^

Note: Same uppercase letter within column represents no significant difference (*p* > 0.05) for within group and between groups, while same lowercase letter within same row represents no significant difference (*p* > 0.05), different symbols within column of Steradent represents significant difference (*p* < 0.05) between groups.

**Table 6 nanomaterials-10-01757-t006:** Colour change values according to NBS unit system after storing the samples for 180 days in different denture cleaners.

Experimental Groups	NBS Values after 180 Days in Cleaners		
	Distilled Water (DW)	Steradent (STD)	Milton (MIL)
G1 (Control 0%)	0.49	1.01*	1.22*
G2 (1.5%)	1.05*	2.76^^^	1.73^^^
G3 (3%)	0.96*	2.84^^^	2.99^^^
G4 (5%)	1.76^^^	3.51^+^	3.10^+^
G5 (7%)	1.78^^^	3.91^+^	4.69^+^
G6 (10%)	1.82^^^	4.42^+^	6.07^++^

Note: Critical marks of colour difference (Table 2): Trace (0.0–0.5); Slight* (0.5–1.5); Noticeable^ (1.5–3.0), Appreciable^+^ (3.0–6.0) and Much^++^ (6.0–12).

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
