# Peer review of "Effect of Cleansers on the Colour Stability of Zirconia Impregnated PMMA Bio-Nanocomposite"

_nanomaterials, 2020, doi:10.3390/nano10091757_

Round 1

Reviewer 1 Report

This study deals with the color change in Zr reinforced PMMA when subjected to denture cleaners. While the study is novel and well-designed, there are a few questions for the authors:

1) Please elaborate on the clinical significance of Zr doped PMMA. What are their indications? I noticed that the Zr doped PMMA on the images are white. Are these already being utilized clinically? If so, how do they mask the white hue? What are other physical properties of this material? How does Zr bond with the PMMA? 

2) The formula for the delta E is one of the older formulas. What is the justification for using this instead of the newer formula?

3) The samples in Figure 1 do not appear flat or even smooth. Were these samples polished at all? Doesn't the surface roughness of the sample have an effect on the delta E value?

4) In the discussion, please explain possible mechanisms for why the Zr infused PMMA produced a greater color change than the controls. Authors cited some mechanisms from other studies but what do the authors theorized in their own study?

5) The authors used the NSB standards for clinical acceptability of delta E as being any value below 3. Can the authors cite clinical color perceptibility studies that validate this value as being clinically acceptable?

Author Response

All changes in the manuscript are highlighted with yellow colour.

Comments:

1) Please elaborate on the clinical significance of Zr doped PMMA. What are their indications?

I noticed that the Zr doped PMMA on the images are white.

Are these already being utilized clinically? If so, how do they mask the white hue?

What are other physical properties of this material?

How does Zr bond with the PMMA?

Responses:

Addition of Zirconia improved the flexural strength that we found in our previous investigation (see Zidan et al 2019). This could improve the clinical service life of denture. This has been mentioned in the last paragraph of the Discussions section.

White hue that can be seen from the images are not the nanoparticles. For example, in control group with Steradent and Milton where there was no zirconia. This could be due to the reaction between the cleaners and the PMMA. This has already been discussed in 7th Paragraph in Discussions and 7th paragraph in the Results sections.

To the best of the author’s knowledge, addition of Zirconia is still at the research stage. Hopefully in near future, we will see its application in clinical practice.

We have measured other physical properties such sorption, solubility and surface roughness. The sorption /solubility results are accepted for publication in Materials MDPI journal (https://www.mdpi.com/1996-1944/13/17/3732/htm).

And Surface roughness results are under review in Journal of Prosthodontic Research. Therefore, we have not discussed those properties in this paper.

Zirconia bonding procedure is briefly mentioned in Section 2.2 with reference. We are currently in the process of submitting another paper in Dental Materials journal detailing the procedure.

Comments:

2) The formula for the delta E is one of the older formulas. What is the justification for using this instead of the newer formula?

Responses:

The formula for delta E that we have used in this paper for colour measurement is widely applied in both research and clinical applications. See latest example references in very high-quality Journals. The first two latest references are also cited in Table 2 and 2nd paragraph of Discussions section of this paper to make it clear.

Sampaio-Fernandes, Margarida, João Galhardo, Sandra Campos, Susana João Oliveira, José Carlos Reis-Campos, Roberto C. Stegun, and Maria Helena Figueiral. "Colour changes of two thermoplastic resins used for flexible partial dentures." Computer Methods in Biomechanics and Biomedical Engineering: Imaging & Visualization (2020): 1-6.

Nagakura, M., Tanimoto, Y., & Nishiyama, N. (2018). Color stability of glass-fiber-reinforced polypropylene for non-metal clasp dentures. Journal of prosthodontic research, 62(1), 31-34.

Hong G, Murata H, Li Y et al. Influence of denture cleansers on the color stability of three types of denture base acrylic resin. J Prosthet Dent. 2009, 101, 205-213

Comments:

3) The samples in Figure 1 do not appear flat or even smooth. Were these samples polished at all? Doesn't the surface roughness of the sample have an effect on the delta E value?

Responses:

The surfaces were polished even though the surface does not appear quite flat particularly in the experimental groups possibly due to deformation caused by immersion under denture cleaners. This has been mentioned in the 8th paragraph in the Discussions section. The polishing procedure is explained in Section 2.2.

Comments:

4) In the discussion, please explain possible mechanisms for why the Zr infused PMMA produced a greater color change than the controls. Authors cited some mechanisms from other studies but what do the authors theorized in their own study?

Responses:

Addition of white color zirconia could be the main reason for the colour change. This has been mentioned in 7th Paragraph in Discussions Section.

Comments:

5) The authors used the NSB standards for clinical acceptability of delta E as being any value below 3. Can the authors cite clinical color perceptibility studies that validate this value as being clinically acceptable?

Responses:

Please follow these two papers where they clearly mentioned that any value under 3.0 is clinically acceptable. We have also cited the references in Table 2 and 2nd paragraph of Discussions section.

Nagakura, M., Tanimoto, Y., & Nishiyama, N. (2018). Color stability of glass-fiber-reinforced polypropylene for non-metal clasp dentures. Journal of prosthodontic research, 62(1), 31-34.

Sampaio-Fernandes, Margarida, João Galhardo, Sandra Campos, Susana João Oliveira, José Carlos Reis-Campos, Roberto C. Stegun, and Maria Helena Figueiral. "Colour changes of two thermoplastic resins used for flexible partial dentures." Computer Methods in Biomechanics and Biomedical Engineering: Imaging & Visualization (2020): 1-6.

Reviewer 2 Report

The topic of this article entitled “Effect of cleansers on the color stability of zirconia impregnated PMMA bio-nanocomposite.” is aimed to assess the relationship between the use of specific cleansers and the color stability of zirconia. It is an interesting topic and within the journal's scope. Nevertheless, this reviewer would suggest some improvements, before further considerations.  The study has certainly new information related to the colour stability of resins loaded with zirconia nanoparticles and treated with denture cleaners. This article also aims to translate the main key-concepts to tissue engineering and translational medicine in the very next future.

The main strength of this paper is related to the appropriateness of approach, and to the study design, which were both really relevant. Moreover, the soundness of the whole article and the relevancy of discussion have merits, as well. The clarity of writing is also fine; however, some key-concepts should be slightly improved for a major clarity to the readers, involving other critical discussion and better comparing some evidence reported in previously published papers.

Introduction is poor: Authors have reported several important topics related to tissue healing and repairing; however, poor has been reported on the role of soft/hard tissue repairing/regenerating, and on the role of resident stem cells, which can act as the immunomodulatory and pro-osteogenic activities in the local environment (Please, see and discuss: Ballini, A.; Scacco, S.; Coletti, D.; Pluchino, S.; Tatullo, M. Mesenchymal Stem Cells as Promoters, Enhancers, and Playmakers of the Translational Regenerative Medicine. Stem Cells Int. 2017;2017:3292810. – and - Ballini, A.;  Cantore, S.;  Scacco, S.;  Coletti,  D.; Tatullo M. Mesenchymal Stem Cells as Promoters, Enhancers, and Playmakers of the Translational Regenerative Medicine 2018. Stem Cells Int. 2018 Oct 30;2018:69274019).

Something more should be discussed about the role of specific “biomaterials” or “scaffolds” as study model. In this light it’s important to briefly describe something about the “safe” in-vitro reparative models, working without any additive (e.g. BSA) to apply safely such protocols in human models (Please, see and discuss “Highly Efficient In Vitro Reparative Behaviour of Dental Pulp Stem Cells Cultured with Standardised Platelet Lysate Supplementation. Stem Cells Int. 2016;2016:7230987.”) Finally, the strategies to chose materials and manage loads should be briefly described in the discussion section, highlighting the role of pre-clinical investigations on this matter (Please, see and discuss “Marrelli M, Maletta C, Inchingolo F, Alfano M, Tatullo M. Three-point bending tests of zirconia core/veneer ceramics for dental restorations. Int J Dent 2013; 2013, 831976.”).

- Authors have reported that “Denture cleaners are commonly employed to remove stains and debris from denture surfaces and to prevent the formation of plaque or colonization of bacteria “ this is a pivotal concept in your paper that must be improved  (Please, see and discuss: “Cantore, S.; Ballini, A.; Mori, G.; Dibello, V.; Marrelli, M.; Mirgaldi, R.; De Vito, D.; Tatullo, M. Anti-plaque and antimicrobial efficiency of different oral rinses in a 3-day plaque accumulation model. J. Biol Regul Homeost Agents 2016, 30, 1173–1178.”)

- Conclusions should be improved with clear take-home messages.

Minor suggestion:

- In the whole text there are some typos here and there: authors should carefully revise the text before resubmission.

Author Response

All changes in the manuscript are highlighted with green colour.

Comments:

The topic of this article entitled “Effect of cleansers on the color stability of zirconia impregnated PMMA bio-nanocomposite.” is aimed to assess the relationship between the use of specific cleansers and the color stability of zirconia. It is an interesting topic and within the journal's scope.

Nevertheless, this reviewer would suggest some improvements, before further considerations. The study has certainly new information related to the colour stability of resins loaded with zirconia nanoparticles and treated with denture cleaners. This article also aims to translate the main key-concepts to tissue engineering and translational medicine in the very next future. The main strength of this paper is related to the appropriateness of approach, and to the study design, which were both really relevant. Moreover, the soundness of the whole article and the relevancy of discussion have merits, as well. The clarity of writing is also fine; however, some key-concepts should be slightly improved for a major clarity to the readers, involving other critical discussion and better comparing some evidence reported in previously published papers.

Responses:

The authors would like to thank you for your positive comments.

Comments:

Introduction is poor: Authors have reported several important topics related to tissue healing and repairing; however, poor has been reported on the role of soft/hard tissue repairing/regenerating, and on the role of resident stem cells, which can act as the immunomodulatory and pro-osteogenic activities in the local environment (Please, see and discuss: Ballini, A.; Scacco, S.; Coletti, D.; Pluchino, S.; Tatullo, M. Mesenchymal Stem Cells as Promoters, Enhancers, and Playmakers of the Translational Regenerative Medicine. Stem Cells Int. 2017;2017:3292810. – and - Ballini, A.; Cantore, S.; Scacco, S.; Coletti, D.; Tatullo M. Mesenchymal Stem Cells as Promoters, Enhancers, and Playmakers of the Translational Regenerative Medicine 2018. Stem Cells Int. 2018 Oct 30;2018:69274019).

Responses:

The authors would like to thank the reviewer for this suggestion. However, we did not mention anything related to tissue healing and repairing. The authors would like to emphasize that in this paper the effect of denture cleaners on nanocomposite color change is the main focus of this paper. After careful reviewing of the suggested papers, we strongly felt that they are not quite relevant to this investigation.

Comments:

Something more should be discussed about the role of specific “biomaterials” or “scaffolds” as study model. In this light it’s  important to briefly describe something about the “safe” in-vitro  reparative models, working without any additive (e.g. BSA) to apply  safely such protocols in human models (Please, see and discuss  “Highly Efficient In Vitro Reparative Behaviour of Dental Pulp Stem  Cells Cultured with Standardised Platelet Lysate Supplementation. Stem Cells Int. 2016; 2016:7230987.”)

Responses:

We would like to thank the reviewer for bringing this to our attention. This has been discussed with the suggested reference in the Discussions section.

Comments:

Finally, the strategies to choose materials and manage loads should be briefly described in the discussion section, highlighting the role of pre-clinical investigations on this matter (Please, see and discuss “Marrelli M, Maletta C, Inchingolo F, Alfano M, Tatullo M. Three-point bending tests of zirconia core/veneer ceramics for dental restorations. Int J Dent 2013; 2013, 831976.”).

Responses:

The role of pre-clinical investigation on the materials selection strategy are explained in the last paragraph of the Discussion section.

Comments:

Authors have reported that “Denture cleaners are commonly  employed to remove stains and debris from denture surfaces and to prevent the formation of plaque or colonization of bacteria “ this  is a pivotal concept in your paper that must be improved (Please, see and discuss: “Cantore, S.; Ballini, A.; Mori, G.; Dibello, V.;  Marrelli, M.; Mirgaldi, R.; De Vito, D.; Tatullo, M. Anti-plaque and  antimicrobial efficiency of different oral rinses in a 3-day plaque  accumulation model. J. Biol Regul Homeost Agents 2016, 30,  1173–1178.”)

Responses:

The authors would like to thank the reviewer for the suggestion. The concept of “Anti-plaque and antimicrobial efficiency” has been further extended with reference in the 2nd paragraph of Introduction section and last paragraph of the discussion section.

Comments:

Conclusions should be improved with clear take-home messages.

Responses:

Conclusions are improved as per the suggestion.

Comments:

Minor suggestion: - In the whole text there are some typos here and there: authors should carefully revise the text before resubmission.

Responses:

Minor typographical errors are corrected and highlighted with green colour throughout the paper.

Comments:

Additional changes

Responses:

References are rearranged in the text and in the list of reference section

Reviewer 3 Report

The authors investigated the effect of cleansers for color changes of PMMA bio-nanocomposite. They focused on denture cleaners and zirconia impregnated PMMA bio-nanocomposite. The paper provides interesting data but I have some concerns about methodology as follows.

P2, Materials and methods

It would be easier to read this article if the authors describe details of the using materials e.g. concentration of sodium hypochlorite, potassium carbonate peroxide, and others (wt%, mass%, or mol%) in the Table.

P3, PMMA specimen

How about specimen polishing? The reviewer and reader want to know polishing disc specimen in detail.

P4, statistical analysis

The authors conducted non-parametric procedure. The reviewer considers the statistical analysis is appropriate.

Author Response

All changes in the manuscript are highlighted with cyan colour.

We would like to thank the reviewer for spending time reviewing the article and providing constructive feedback. Having read the comments and consulted with the co-authors we have now made the appropriate amendments. We hope that the revised version meets your approval.

Comments:

The authors investigated the effect of cleansers for color changes of PMMA bio-nanocomposite. They focused on denture cleaners and zirconia impregnated PMMA bio-nanocomposite. The paper provides interesting data but I have some concerns about methodology as follows.

Responses:

The authors would like to thank you for your positive comments.

Comments:

P2, Materials and methods It would be easier to read this article if the authors describe details of the using materials e.g. concentration of sodium hypochlorite, potassium carbonate peroxide, and others (wt%, mass%, or mol%) in the Table.

Responses:

We completely agree with the reviewer. Unfortunately, the concentrations of the chemicals are not known from the manufacturer.

Comments:

P3, PMMA specimen How about specimen polishing? The reviewer and reader want to know polishing disc specimen in detail.

Responses:

The polishing procedure is explained in Section 2.2.

Comments:

P4, statistical analysis The authors conducted non-parametric procedure. The reviewer considers the statistical analysis is appropriate.

Responses:

Shapiro-Wilk test showed that the data were not normally distributed. Therefore, the authors conducted the non-parametric procedure.

Round 2

Reviewer 1 Report

Queries have been responded to sufficiently.

Reviewer 2 Report

authors have well addressed the comments